# An ancient metabolite damage-repair system sustains photosynthesis in plants

Dario Leister [1], Anurag Sharma [2,3], Natalia Kerber[1], Thomas Nägele[4], Bennet Reiter [1], Viviana Pasch [1], Simon Beeh [1,5], Peter Jahns[6], Roberto Barbato [7], Mathias Pribil[3] & Thilo Rühle [1]✉

Ribulose-1,5-bisphosphate carboxylase/oxygenase (Rubisco) is the major catalyst in the conversion of carbon dioxide into organic compounds in photosynthetic organisms. However, its activity is impaired by binding of inhibitory sugars such as xylulose-1,5-bisphosphate (XuBP), which must be detached from the active sites by Rubisco activase. Here, we show that loss of two phosphatases in *Arabidopsis thaliana* has detrimental effects on plant growth and photosynthesis and that this effect could be reversed by introducing the XuBP phosphatase from *Rhodobacter sphaeroides*. Biochemical analyses revealed that the plant enzymes specifically dephosphorylate XuBP, thus allowing xylulose-5-phosphate to enter the Calvin-Benson-Bassham cycle. Our findings demonstrate the physiological importance of an ancient metabolite damage-repair system in degradation of by-products of Rubisco, and will impact efforts to optimize carbon fixation in photosynthetic organisms.

The catalytic conversion of atmospheric $CO_2$ into carbohydrates in photosynthetic organisms is largely mediated by a single enzyme – ribulose-1,5-bisphosphate carboxylase/oxygenase (Rubisco). It is estimated that Rubisco is the most abundant protein on Earth[1] and fixes about 105 petagrams ($10.5^{11}$ tons) of $CO_2$ annually[2]. Carboxylation of ribulose-1,5-bisphosphate (RuBP) is the first step in the Calvin-Benson-Bassham (CBB) cycle, which requires the activation of Rubisco through carbamylation of a conserved lysine residue in the active site by a non-substrate $CO_2$ molecule[3]. The carbamylated lysine is then stabilized by the subsequent binding of a magnesium ion, which enables the efficient electrophilic attack of RuBP by the substrate $CO_2$ molecule. Despite its central role in the CBB cycle, Rubisco has a low turnover rate of 3-10 $CO_2$ molecules per sec[4] and its complex reaction mechanism is prone to error. One of its by-products – 2-phosphoglycolate[5] – effectively inhibits several enzymes of primary carbon metabolism in photosynthetic organisms, including triose-phosphate isomerase[6], phosphofructokinase[7] and sedoheptulose 1,7-bisphosphate phosphatase[8]. In addition, 2-phosphoglycolate is recycled in a metabolic process called photorespiration which requires ATP and multiple enzymes, and involves no less than four subcellular compartments[9]. Another limitation for efficient carboxylation is that protonation and oxygenation of the RuBP enediolate intermediate gives rise to several isomeric pentulose bisphosphates [2,3-pentodiulose-1,5-bisphosphate (PDBP), 3-ketoarabinitol-1,5-bisphosphate and xylulose-1,5-bisphosphate (XuBP)][10], which are also produced during in-vitro studies with isolated Rubisco complexes. In such assays, the effects of these compounds become manifest in the progressive inactivation of Rubisco, which is referred to as 'fallover'[11]. RuBP itself can also act as an inhibitory component when it binds to the uncarbamylated form of the enzyme[12]. Moreover, some plant species produce 2-carboxy-D-arabinitol 1-phosphate (CA1P), which accumulates in the dark or under low light[13] and binds only to carbamylated Rubisco[14]. In contrast to other sugar phosphate inhibitors, CA1P is derived from fructose-1,6-bisphosphate (FuBP) and is synthesized via a pathway that

[1]Plant Molecular Biology, Faculty of Biology, Ludwig-Maximilians-University Munich, D-82152 Planegg-Martinsried, Germany. [2]Electron Microscopy Resource Center, The Rockefeller University, New York, NY, USA. [3]Copenhagen Plant Science Centre, Department of Plant and Environmental Sciences, University of Copenhagen, 1871 Copenhagen, Denmark. [4]Plant Evolutionary Cell Biology, Faculty of Biology, Ludwig-Maximilians-University Munich, D-82152 Planegg-Martinsried, Germany. [5]Department of Plant Physiology, Centre of Plant Molecular Biology (ZMBP), University of Tübingen, Tübingen, Germany. [6]Plant Biochemistry, Heinrich-Heine-University Düsseldorf, D-40225 Düsseldorf, Germany. [7]Dipartimento di Scienze e Innovazione Tecnologica, Università del Piemonte Orientale, 15121 Alessandria, Italy. ✉e-mail: thilo.ruehle@biologie.uni-muenchen.de

is comprised of five reactions[15]. All of these inhibitors are structurally similar to transitory forms of RuBP that arise during catalysis and all bind to the enzyme's active site, thus locking Rubisco into the closed, catalytically impaired complex.

The release of such inhibitors is in turn achieved by the action of the AAA+ (ATPase Associated with diverse cellular Activities) Rubisco activase (Rca), which facilitates conformational changes that convert Rubisco into a catalytically competent complex once again[16–18]. Although Rca enables the release of inhibitors, they must then be rapidly degraded to prevent them from rebinding to Rubisco. For instance, the inhibitor CA1P, which accumulates in the dark, is converted into the non-inhibitory component 2-carboxy-D-arabinitol (CA) by the phosphatase CA1Pase[19].

Recently, another neutralising phosphatase has been identified, which is specific for the inhibitory sugar XuBP[20]. XuBP is generated by misprotonation of the 2,3-enediol derivative of RuBP[21] and binds to both the decarbamylated and the carbamylated form of Rubisco[22]. In plants, rates of synthesis of XuBP are much higher than those of other sugar inhibitors[23] and increase with temperature[24,25]. XuBP is a competitive inhibitor of RuBP and is also a poor substrate for the carboxylation reaction catalyzed by Rubisco[26]. XuBP phosphatases (CbbY) are conserved in plants and algae, as well as in many autotrophic bacteria, in which CbbY is encoded in the *cbb* operon (the Calvin-Benson-Bassham cycle operon). A combination of biochemical and structural analyses has revealed that CbbY proteins from *Rhodobacter sphaeroides* (RsCbbY) and its conserved counterpart in *Arabidopsis thaliana* (AtCbbYA) are highly specific XuBP phosphatases and transform XuBP into the harmless CBB cycle intermediate xylulose 5-phosphate (Xu5P), which can be recycled back into RuBP[20].

In this study, we demonstrate that an ancient metabolite-repair system is required to maintain photosynthesis in plants. In addition, we describe a second metabolite damage-repair enzyme in Arabidopsis chloroplasts, which is able to recycle XuBP. XuBP dephosphorylation activity turns out to be critical for efficient carbon fixation. As shown here, its loss impairs photosynthesis and plant growth. Moreover, we demonstrate that the metabolite-damage repair system found in purple bacteria is compatible with that of land plants, as the introduction of the XuBP phosphatase from *Rhodobacter sphaeroides* complemented Arabidopsis plants that lacked their own XuBP phosphatases.

## Results

### The haloacid dehalogenase-like hydrolase AtCbbYB shares sequence similarity with AtCbbYA

In a screen for novel photosynthesis-related proteins in Arabidopsis, we focused on candidates that are (i) shared by photosynthetic eukaryotes from the green lineage and (ii) display a photosynthesis-specific mRNA expression profile. To this end, functional profiling analysis[27] of Arabidopsis genes[28] that were candidates from the GreenCut2 collection[29] was carried out. One of the candidates that was co-expressed with genes (Supplementary Data 1) which exhibited a significant degree of enrichment (adjusted *p*-value of $4.3 \times 10^{-12}$) for the functional category 'photosynthesis' (gene ontology term GO:0015979) was *AT3G48420* (Supplementary Data 2), which is referred to as Arabidopsis *CBBYA* or *AtCBBYA* in the following. AtCbbYA belongs to the large class of haloacid dehalogenase-like hydrolases (HAD)[30] and shares similarity with the protein product of *AT4G39970.1*, which is designated as AtCbbYB. A phylogenetic analysis revealed that homologues of AtCbbYB are found in green algae and land plants (Supplementary Fig. 1a, b, Supplementary Table 1). The 3D model of AtCbbYB (Fig. 1a) predicted by AlphaFold[31] has a bipartite structure comprised of a core and a cap domain, which is also found in RsCbbY and AtCbbYA[20], as well as in other members of the HAD family. Sequence alignment of AtCbbYA, AtCbbYB and RsCbbY (Fig. 1b) revealed that most of the catalytic (7 out of 8) and several signature

residues (7 out of 13) defined by analyses of the crystal structure of RsCbbY bound to XuBP[20] are conserved in AtCbbYB.

### Loss-of-function mutations in AtCbbYA and AtCbbYB impair plant growth

Arabidopsis T-DNA insertion lines for *AtCBBYA* (*atcbbya*, Salk Institute line SALK_025204) and *AtCBBYB* (*atcbbyb*, John Innes Centre line SM_3.15346) were identified, and the double mutant line *atcbbyab* was generated by crossing the two single mutants (Fig. 2a). Northern analyses indicated that the *AtCBBYA* mRNA was truncated in both the *atcbbya* and *atcbbyab* lines (Fig. 2b), whereas the *AtCBBYB* transcript was undetectable in *atcbbyb* and *atcbbyab* (Fig. 2c). To analyze the impact of the gene disruptions at the protein level (Fig. 2d), antibodies were raised against recombinantly expressed and purified AtCbbYA and AtCbbYB. Immunodetection assays showed that *atcbbya* and the double mutant *atcbbyab* failed to accumulate AtCbbYA, whereas AtCbbYA was still present in *atcbbyb* (Fig. 2d). Conversely, AtCbbYB could not be detected in either *atcbbyb* or the double mutant, but accumulated in *atcbbya*. No obvious growth phenotype was observed for *atcbbya* or *atcbbyb*, but growth of the double mutant *atcbbyab* was clearly reduced relative to wild-type plants under all conditions tested (Fig. 2e, f). We confirmed that the growth phenotype observed in *atcbbyab* resulted from the knockout of both genes by complementing the *atcbbyab* mutation with either the *AtCBBYA* or *AtCBBYB* gene fused to the eGFP-encoding reporter gene and placed under the control of the Cauliflower Mosaic Virus (CaMV) 35S promoter (Supplementary Fig. 2a–c). Transformed plants bearing eGFP fusion constructs were also employed to determine the subcellular localization of AtCbbYA and AtCbbYB (Supplementary Fig. 3). Both were targeted to the chloroplast, as demonstrated by fluorescence microscopy of isolated protoplasts (Supplementary Fig. 3a). Furthermore, stromal localization was confirmed by fractionation experiments carried out on wild-type plants, followed by immunodetection of AtCbbYA or AtCbbYB (Supplementary Fig. 3b). The fact that plant growth was visibly perturbed only when both proteins were absent indicates that AtCbbYA and AtCbbYB are functionally equivalent with respect to metabolite damage-repair of Rubisco during photosynthesis.

### AtCbbYA and AtCbbYB are required for efficient photosynthetic electron transport and CO₂ assimilation

The detrimental effects of simultaneous loss of AtCbbYA and AtCbbYB on plant growth prompted us to explore the influence of the two proteins on the light reactions of photosynthesis in more detail (Fig. 3). The maximum quantum yield of PSII (Fv/Fm) as determined in the dark-adapted state was slightly lower (Fig. 3a), whereas the effective quantum yield [Y(II)] at 200 µmol photons $m^{-2}$ $s^{-1}$ was reduced to about 50% of wild-type levels in *atcbbyab* (Fig. 3b). The reduction in PSII quantum efficiency in *atcbbyab* was shown to be attributable to a significant increase in the yield of regulated, non-photochemical quenching [Y(NPQ)] (Fig. 3c), while the level of non-regulated energy dissipation [Y(NO)] was unaltered (Fig. 3d). We also identified an increase in PSII excitation pressure (1-qP), indicating a higher reduction state of the plastoquinone pool in the double mutant relative to the wild type (Fig. 3e). A pronounced decrease in the quantum yield [Y(I)] of PSI was detected only in the double mutant (Fig. 3f). This effect could be ascribed to a sharp rise in the donor-side limitation [Y(ND)] of PSI (Fig. 3g), whereas the acceptor side limitation [Y(NA)] in the double mutant was slightly lower than in the wild type (Fig. 3h). As expected on the basis of their normal growth phenotype (Fig. 2f), the single mutants behaved like the wild type with respect to all photosynthetic parameters (Fig. 3). Moreover, the aberrant Y(II) and Y(NPQ) phenotypes of *atcbbyab* were corrected in the complementation lines $P_{35S}$:*AtCbbYA-eGFP* and $P_{35S}$:*AtCbbYB-eGFP*, as was confirmed by chlorophyll *a* fluorescence video imaging analyses (Supplementary Fig. 2).

We further investigated the pigment composition of *atcbby* mutants by high-performance liquid chromatography (Supplementary

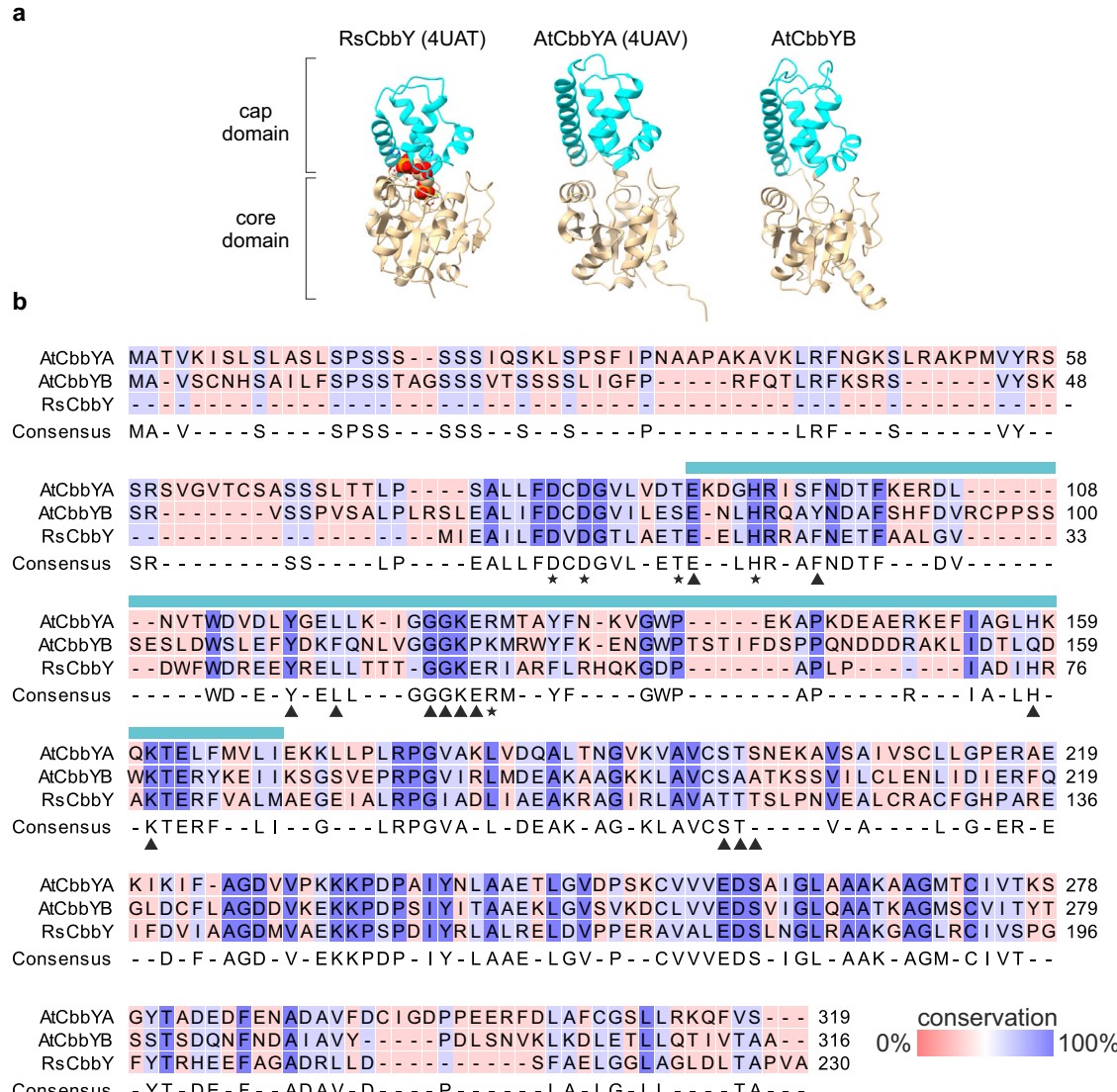

**Fig. 1 | Protein structure and sequence alignment of CbbY proteins. a** Structures of CbbY from *Rhodobacter sphaeroides* (RsCbbY, 4UAT) and CbbYA from *Arabidopsis thaliana* CbbYA (AtCbbYA, 4UAV) were recently resolved[20], and the structure of CbbYB from *Arabidopsis thaliana* CbbYB (AtCbbYB) was predicted by AlphaFold[31]. RsCbbY is shown in a complex with XuBP and Mg²⁺, both of which are highlighted in spherical shapes. Note that the predicted transit peptide (1-63 AA) of AtCbbYB was excluded from the 3D model. **b** Sequence alignment of At3g48420.1 (AtCbbYA), At4g39970.1 (AtCbbYB) and RsCbbY. Conservation is depicted on a color scale from red (0%) to blue (100%). The positions of catalytic and signature residues conserved in AtCbbYA and RsCbbY[20] are depicted in black stars and triangles, respectively. The cap domain is indicated by a turquoise bar above the sequences.

Table 2). Levels of chlorophylls *a* and *b*, as well as total carotenoids were unaltered in all *atcbby* mutants. However, the pool of violaxanthin, antheraxanthin and zeaxanthin was clearly increased in *atcbbyab* (72.2 ± 9.3 pmol mg⁻¹ FW) with respect to the wild type (40.2 ± 4.6 pmol mg⁻¹ FW), *atcbbya* (45.2 ± 5.9 pmol mg⁻¹ FW) and *atcbbyb* (43.4 ± 6.3 pmol mg⁻¹ FW). Indeed, antheraxanthin levels in *atcbbyab* (12.4 ± 1.3 pmol mg⁻¹ FW) were about eight-fold higher than in the wild type (1.6 ± 0.5 pmol mg⁻¹ FW), while zeaxanthin reached detectable levels only in the double mutant (8.4 ± 0.9 pmol mg⁻¹ FW).

To quantify the impact of the loss of AtCbbYA and/or AtCbbYB function on CBB cycle activity, we performed gas-exchange measurements while simultaneously monitoring Chl *a* fluorescence (Fig. 4). First, we determined the response of carbon assimilation rates to photon flux (Fig. 4a) and included in our analysis the Rubisco small subunit (RbcS) knockdown line *rbcs1a3b-1* as a control for reduced CBB cycle activity[32]. CO₂ assimilation rates at saturating light intensities of 1200 μmol photons m⁻² s⁻¹ (Fig. 4a) were lower in *atcbbyab* (141 ± 23 μmol CO₂ g⁻¹ FW h⁻¹) than in wild type (302 ± 49 μmol CO₂ g⁻¹

FW h⁻¹), but not as severely impaired as in the control line *rbcs1a3b-1* (79 ± 11 μmol CO₂ g⁻¹ FW h⁻¹). As expected, PSII electron transport rates displayed identical trends (Fig. 4b), since the quantum yield of CO₂ assimilation and the photochemical yield of PSII have been shown to correlate over a wide range of light intensities[33]. We also confirmed the high non-photochemical quenching phenotype of *atcbbyab* (Fig. 3c) and identified an even more pronounced Y(NPQ) phenotype for the RbcS knockdown line *rbcs1a3b-1*, which was particularly evident at light intensities of 300 and 600 μmol photons m⁻² s⁻¹ (Fig. 4c). Increasing the CO₂ concentration under saturating light conditions (1200 μmol photons m⁻² s⁻¹) could not restore *atcbbyab* and *rbcs1a3b-1* assimilation rates to wild-type levels (Fig. 4d). Analyses of gas-exchange kinetics and Chl *a* fluorescence in the single mutants *atcbbya* and *atcbbyb* did not reveal any obvious differences from the wild type (Supplementary Fig. 4a–c). Furthermore, the absence of either AtCbbYA or AtCbbYB or both had no effects on the level of either Rubisco or Rubisco activase, as demonstrated by immunodetection assays (Supplementary Fig. 4d). Based on these results, we deduced

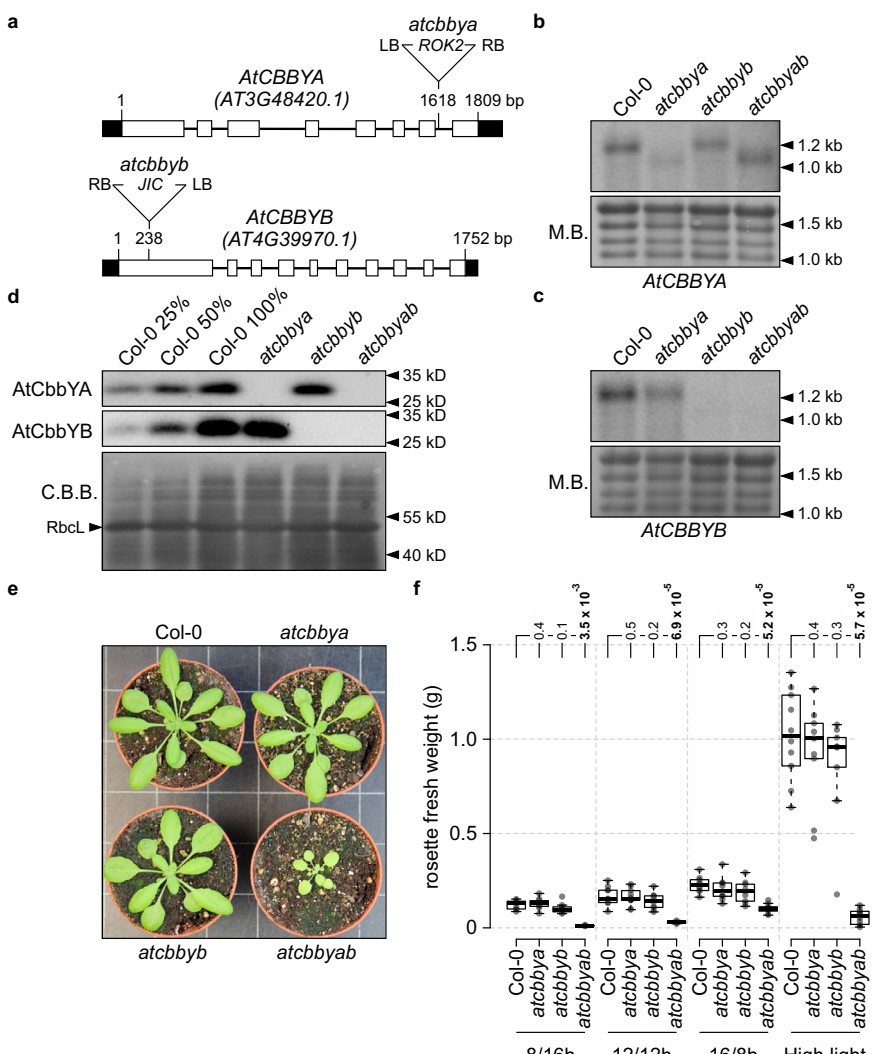

**Fig. 2 | Generation of the double mutant *atcbbyab*. a** Exon-intron structures of the *AtCBBYA* (*AT3G48420.1*) and *AtCBBYB* (*AT4G39970.1*) genes and sites of insertions in the respective T-DNA lines. Left and right T-DNA borders are labelled by "LB" and "RB", respectively. Exons are shown as white rectangles, UTRs as black rectangles. Positions of T-DNA integrations are highlighted. **b, c** Northern analyses of *AtCBBYA* (**b**) and *AtCBBYB* (**c**) transcripts. Total RNA of Col-0, *atcbbya*, *atcbbyb* and *atcbbyab* was extracted, size-fractionated on denaturing agarose gels, transferred onto nylon membranes and examined by hybridization of radiolabeled *AtCBBYA*- and *AtCBBYB*-specific probes. As loading controls, abundant rRNAs were visualized by methylene blue staining (M.B.). Northern analyses were carried out once. **d** Immunodetection of AtCbbYA and AtCbbYB by Western analyses. A Coomassie-stained gel (C.B.B.) bearing size-fractionated proteins extracted from total leaves is shown as loading control. Immunodetections were repeated twice with similar results. **e** Col-0, *atcbbya*, *atcbbyb* and *atcbbyab* plants grown under a 12/12 h day/night cycle (80–100 μmol photons m⁻² s⁻¹) and photographed 26 days after germination. **f** Growth measurements. Rosette fresh weight was determined from plants grown under the indicated light regimes [8/16 h short day (80–100 μmol photons m⁻² s⁻¹), 12/12 h (80–100 μmol photons m⁻² s⁻¹), 16/8 h long day (80–100 μmol photons m⁻² s⁻¹), and under high light levels (~400 μmol photons m⁻² s⁻¹ on a 12/12 h day/night cycle) in the presence of fertilizer]. Ten plants per genotype (9 *atcbbyb* plants under long day, and 9 *atcbbya* plants under high light conditions) were examined. Centre lines show the medians, and boxes indicate the 25th and 75th percentiles. Whiskers denote 1.5x the interquartile range. Data points are plotted as circles and outliers are represented by dots. For statistical analyses, the non-parametric Kruskal–Wallis test was performed, followed by pairwise Dunn's tests. The *p*-values were adjusted on an experiment level using the Benjamini–Hochberg method. *P*-values for comparison of each line with Col-0 are indicated and $p \leq 0.05$ are marked in bold.

that both AtCbbYA and AtCbbYB are required for the maintenance of efficient photosynthetic electron transport and carbon dioxide assimilation. In contrast to *rbcs1a3b-1*, in which decreased $CO_2$ assimilation rates were correlated with reduced Rubisco content[32], the data for *atcbbyab* indicated that lower rates of carbon fixation could be due to inefficient degradation of inhibitors of Rubisco in the absence of both AtCbbYA and AtCbbYB.

## AtCbbYA and AtCbbYB dephosphorylate XuBP

Since AtCbbYA had been shown to dephosphorylate XuBP[20], we investigated the substrate specificity and kinetic behaviour of AtCbbYB. Fusions of AtCbbYA and AtCbbYB to the maltose-binding protein

(MBP) were heterologously expressed in *E. coli* and subsequently purified under native conditions with the aid of an amylose matrix (Supplementary Fig. 5). Both fusion proteins released one molecule of phosphate per molecule of XuBP and RuBP, but were unable to dephosphorylate either xylulose-5-phosphate (Xu5P) or ribulose-5-phosphate (Ru5P) (Fig. 5a). In contrast, calf intestinal alkaline phosphatase (CIP), with its broad substrate specificity, liberated two phosphates per molecule of XuBP or RuBP, and was also able to dephosphorylate both Xu5P and Ru5P. These results indicated that MBP-AtCbbYB specifically cleaves the phosphate at position 1 in XuBP and RuBP, as has previously been demonstrated for AtCbbYA[20]. Kinetic analyses with XuBP as substrate revealed that the Michaelis-Menten

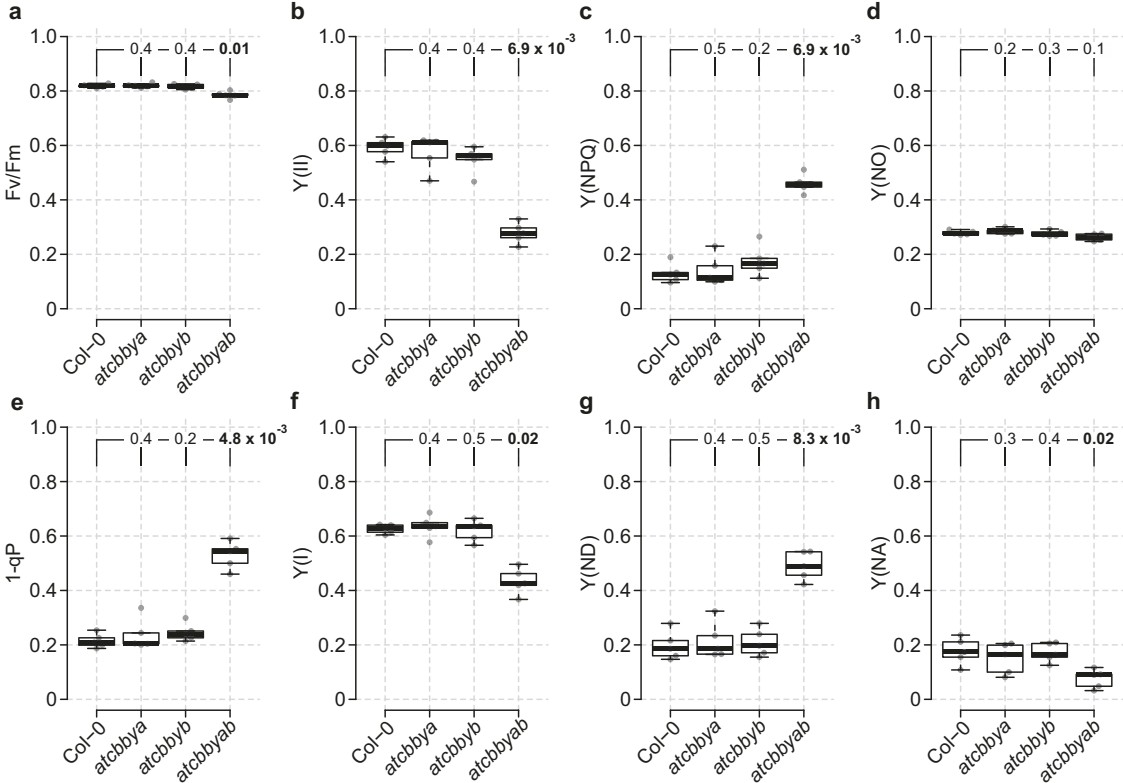

**Fig. 3 | Loss of AtCbbYA and AtCbbYB impairs photosynthetic electron transport.** Chlorophyll *a* fluorescence and P700 parameters of WT (Col-0), *atcbbya*, *atcbbyb* and *atcbbyab* plants grown under long-day conditions at 100 µmol photons m$^{-2}$s$^{-1}$. Plants were dark-adapted for 20 min and single leaves were exposed to 200 µmol photons m$^{-2}$s$^{-1}$ for 10 min. At the end of the actinic light treatment, a saturating pulse was applied and photosynthetic parameters were recorded. **a** Maximum quantum yield of PSII [Fv/Fm]. **b** Effective quantum yield of PSII [Y(II)]. **c** Regulated energy dissipation in PSII [Y(NPQ)], **d** Non-regulated energy dissipation in PSII [Y(NO)], **e** Reduction state of plastoquinone or excitation pressure [1-qP]. **f** Quantum yield of PSI [Y(I)]. **g** Donor-side limitation of PSI [Y(ND)]. **h** Acceptor-side limitation of PSI [Y(NA)]. Horizontal lines represent the medians, and boxes indicate the 25th and 75th percentiles. Whiskers denote 1.5x the interquartile range, outliers are represented by dots. Data points ($n = 5$) are plotted as open circles. For statistical analyses, the non-parametric Kruskal–Wallis test was performed, followed by pairwise Dunn's tests. The *p*-values were adjusted on an experiment level using the Benjamini-Hochberg method. *P*-values for comparison of each line with Col-0 are indicated and $p \leq 0.05$ are marked in bold.

constant ($K_M$) for MBP-AtCbbYB ($0.182 \pm 0.03$ mM) was ~4.5-fold higher than for MBP-AtCbbYA ($0.04 \pm 0.001$ mM) (Fig. 5b, Table 1). Moreover, the turnover number ($k_{cat}$) for MBP-AtCbbYB ($0.54 \pm 0.05$ s$^{-1}$) was ~4 times lower than that of MBP-AtCbbYA ($2.13 \pm 0.04$ s$^{-1}$). Analyses with RuBP as substrate revealed similar $K_M$ values for MBP-AtCbbYA ($4.31 \pm 0.11$ mM) and MBP-AtCbbYB ($4.84 \pm 0.36$ mM) (Fig. 5b, Table 1). Turnover numbers for RuBP relative to XuBP were ~50- and ~35-fold lower for MBP-AtCbbYA ($0.043 \pm 0.009$ s$^{-1}$) and MBP-AtCbbYB ($0.015 \pm 0.002$ s$^{-1}$), respectively, indicating that both phosphatases act selectively on XuBP. Consequently, MBP-AtCbbYA and MBP-AtCbbYB exhibited ~5000 and ~1000 higher catalytic efficiency ($k_{cat}/K_M$), respectively, with XuBP than RuBP.

Overall, our biochemical analyses of MBP-AtCbbYA reproduced previously published data based on untagged AtCbbYA[20] and demonstrated that Arabidopsis contains a second phosphatase that has the ability to degrade the Rubisco inhibitor XuBP to Xu5P. Both phosphatases displayed physiologically relevant high selectivity for XuBP, thus preventing dephosphorylation of the isomeric Rubisco substrate RuBP and subsequent impairment of the carboxylation reaction in the CBB cycle.

### The XuBP phosphatase from *Rhodobacter sphaeroides* can substitute for AtCbbYA and AtCbbYB

Since CbbY homologues have been identified in plants and algae, as well as in many autotrophic bacteria (Supplementary Fig. 1a), we investigated the functional compatibility of the XuBP phosphatase found in *Rhodobacter sphaeroides* (RsCbbY) with those of Arabidopsis. To this end, the codon usage of the *RsCbbY* gene was optimized for efficient expression in Arabidopsis, and fused to the sequence coding for the predicted plastid transit peptide of AtCbbYA. Moreover, the construct was placed under the control of the cauliflower mosaic virus 35S promoter and fused to the eGFP-encoding reporter gene. After introduction of the overexpressor construct into the *atcbbyab* background, two independent lines (*P$_{35S}$:TP-RsCbbY-eGFP #1* and *#2*) were selected. The presence of the fusion was verified by immunodetection assays (Fig. 6a), and localization of TP-RsCbbY-eGFP to the chloroplasts was confirmed by fluorescence microscopy of isolated protoplasts (Fig. 6b). Remarkably, *P$_{35S}$:TP-RsCbbY-eGFP* plants complemented the mutant photosynthetic phenotype, as indicated by the restoration of wild-type-like Y(II) and Y(NPQ) parameters during light induction experiments at 185 µmol photons m$^{-2}$ s$^{-1}$ (Fig. 6c).

## Discussion

Metabolite damage is a widespread phenomenon that occurs in all living organisms and has been studied in more detail in several metabolic contexts, such as glycolysis[34], amino-acid synthesis[35] and damage to phosphometabolites[36]. In particular, plants that have to cope with rapidly changing and extreme environmental conditions are expected to be more susceptible to metabolite damage, and it is estimated that large numbers of uncharacterized genes are associated with the repair of metabolite damage[37]. Although the inhibitory effect of XuBP on

Rubisco activity has long been known[38], the metabolite-repair mechanism has been elucidated at the biochemical level only recently[20]. In our study, we reveal now its physiological importance in plant photosynthesis which has remained unrecognized until now due

to redundant functions of the two XuBP phosphatases. Taken together, the identification of a second XuBP phosphatase in Arabidopsis (Fig. 1) and the pronounced phenotype of the *atcbbyab* mutant underline the significance of functional cooperation between Rubisco activase and sugar phosphatases for the maintenance of normal growth yields (Fig. 2e, f) and photosynthesis (Figs. 3, 4). Both findings demonstrate in planta that the release of XuBP from Rubisco, mediated by Rubisco activase, is insufficient to keep photosynthesis running efficiently, unless the CbbYA/CbbYB degradation system is available to prevent the inhibitor from rebinding (Fig. 7).

Interestingly, *atcbbyab* plants that were grown under high levels of light (~400 μmol photons m$^{-2}$ s$^{-1}$) were much smaller than the wild type (Fig. 2f), This may be due to the increased accumulation of XuBP that occurs under conditions in which carbon assimilation rates are enhanced. However, photosynthesis was already affected in *atcbbyab* plants that were exposed to moderate light intensities (~100 μmol photons m$^{-2}$ s$^{-1}$; Fig. 3) and this could be ascribed to the reduced activity of the CBB cycle (Fig. 4), and the ensuing limited availability of both ADP and the final electron acceptor NADP$^+$ for light reactions (Fig. 7). As a consequence, the trans-thylakoid proton gradient was enhanced in *atcbbyab*, and ΔpH-dependent quenching mechanisms were triggered, which then led to increased xanthophyll cycle activity (Supplementary Table 2) and regulated heat dissipation in the PSII antenna (Fig. 3c). In addition, increased luminal acidification down-regulated plastoquinol oxidation in the Q cycle of the Cyt $b_6f$ complex in the double mutant (Fig. 3e). This mechanism was previously referred to as 'photosynthetic control'[39,40], since it prevents excessive photosynthetic electron transport to PSI. In fact, PSI photochemistry was donor-side-limited in *atcbbyab* (Fig. 3g) and over-reduction at the acceptor side of PSI was alleviated (Fig. 3h). Accordingly, the photosynthetic apparatus was efficiently protected from photodamage by downregulation of linear electron transport in *atcbbyab*, as shown by the fact that maximal quantum yields (Fv/Fm) of PSII remain close to wild-type levels (Fig. 3a).

Although both AtCbbYA and AtCbbYB can dephosphorylate XuBP to Xu5P (Fig. 5) and substitute for each other in complementation experiments (Supplemental Fig. 2), the question remains as to why two different XuBP phosphatases evolved in the green lineage (Supplemental Fig. 1). One possible explanation is that AtCbbYB can dephosphorylate sugar phosphates other than XuBP that inhibit Rubisco

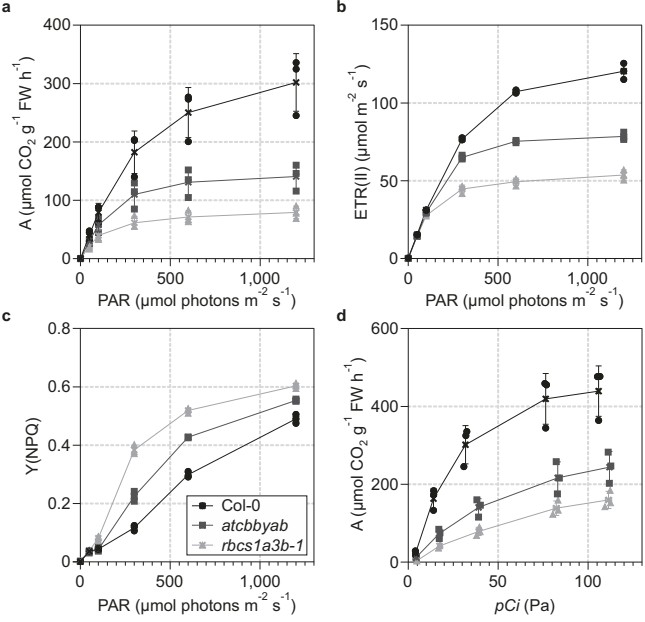

**Fig. 4 | Efficient carbon fixation depends on AtCbbYA and AtCbbYB.** CO$_2$ assimilation and chlorophyll *a* fluorescence of wild type, *atcbbyab* and the Rubisco knockdown mutant *rbcs1a3b-1* were studied using the gas-exchange and fluorescence system GFS-3000 (Walz®, Effeltrich, Germany). **a** CO$_2$ assimilation rates were determined at various light intensities (0, 50, 100, 300, 600 and 1200 μmol photons m$^{-2}$ s$^{-1}$) while the ambient CO$_2$ concentration in the measuring chamber was kept constant (46 Pa). **b, c** Electron transport rates through PSII [ETR(II)] and non-photochemical quenching [Y(NPQ)] parameters were recorded simultaneously with CO$_2$ assimilation rates (**a**). **d** CO$_2$ assimilation rates plotted as a function of increasing intercellular CO$_2$ mole fraction [*pCi*] were determined at a saturating light intensity of 1200 μmol photons m$^{-2}$ s$^{-1}$. Three individual plants per genotype were analyzed. Averages (crosses) and standard deviations are provided.

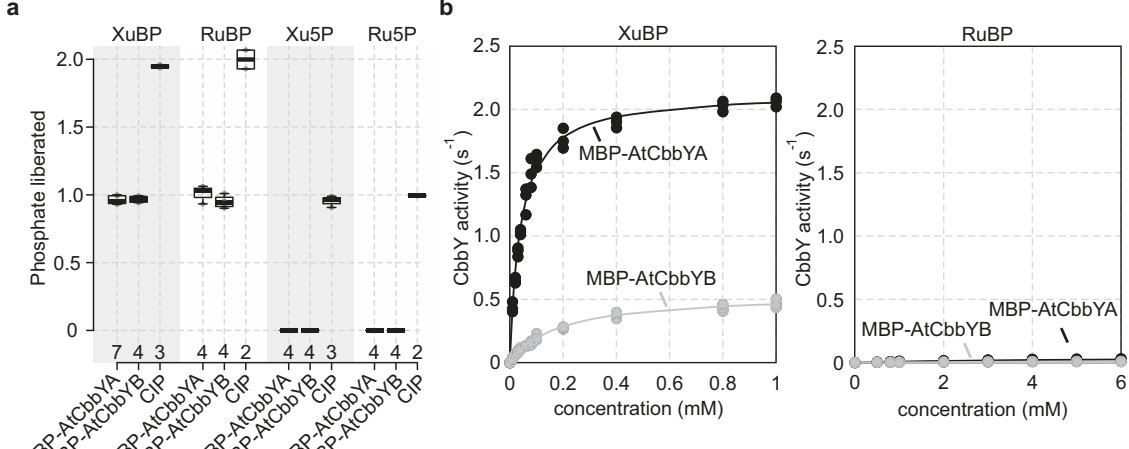

**Fig. 5 | MBP-AtCbbYA and MBP-AtCbbYB specifically dephosphorylate xylulose-1,5-bisphosphate (XuBP). a** Substrate specificity of MBP-AtCbbYA and MBP-AtCbbYB. Calf intestinal phosphatase (CIP) served as a control. Phosphate release from the substrates XuBP, RuBP, Xu5P, and Ru5P was quantified using colorimetric malachite-green assays. Horizontal lines represent the medians, and boxes indicate the 25th and 75th percentiles. Whiskers denote 1.5x the interquartile range. Numbers of replicates are indicated. **b** Michaelis-Menten plots for MBP-AtCbbYA and MBP-AtCbbYB with XuBP and RuBP as substrates. Three measurements were conducted in each experiment. Note that a plot of CbbY kinetics with RuBP as substrate and a fitted scaling of the y-axis is provided in Supplementary Fig. 5b.

**Table 1 | Enzymatic properties of MBP-AtCbbYA and MBP-AtCbbYB**

| | XuBP | | | RuBP | | | $S_{XuBP/RuBP}$ |
|---|---|---|---|---|---|---|---|
| | $k_{cat}$ (s$^{-1}$) | $K_M$ (mM) | $k_{cat}/K_M$ (s$^{-1}$ mM$^{-1}$) | $k_{cat}$ (s$^{-1}$) | $K_M$ (mM) | $k_{cat}/K_M$ (s$^{-1}$ mM$^{-1}$) | |
| MBP-AtCbbYA | 2.13 ± 0.04 | 0.04 ± 0.001 | 53 ± 2 | 0.043 ± 0.009 | 4.31 ± 0.11 | 0.01 ± 0.002 | 5362 ± 852 |
| MBP-AtCbbYB | 0.54 ± 0.05 | 0.182 ± 0.03 | 3 ± 0 | 0.015 ± 0.002 | 4.84 ± 0.36 | 0.0032 ± 0.0004 | 947 ± 676 |

Michaelis–Menten kinetic parameters were determined from three measurements. Means ± standard deviations are provided.

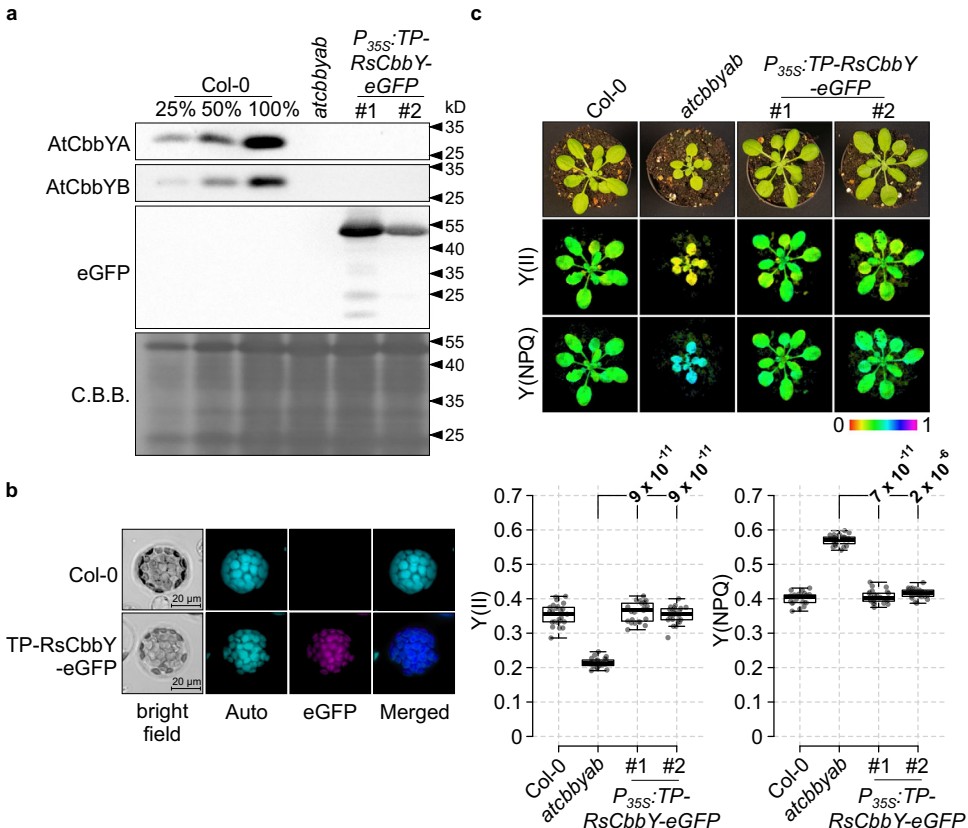

**Fig. 6 | When targeted to chloroplasts, the XuBP phosphatase of *Rhodobacter sphaeroides* (RsCbbY) is functional in Arabidopsis. a** Immunodetection of TP-RsCbbY-eGFP using an eGFP-specific antibody. The absence of AtCbbYA and AtCbbYB in the two $P_{35S}$:*TP-RsCbbY-eGFP* lines was confirmed using antibodies generated in this study. The PVDF membrane stained with Coomassie brilliant blue G-250 (C.B.B.) served as a loading control. Immunodetections were repeated twice with similar results. **b** Localization of TP-RsCbbY-eGFP to chloroplasts was verified by fluorescence microscopy. The integrity of the protoplasts was assessed by bright-field microscopy and chloroplasts were detected by chlorophyll fluorescence (Auto). Fluorescence signals derived from TP-RsCbbY-eGFP (eGFP) were merged with chloroplast-derived chlorophyll fluorescence signals (Merged). **c** Photosynthetic activity in Col-0, *atcbbyab* and two independent $P_{35S}$:*TP-RsCbbY-eGFP* overexpressor lines (*atcbbyab* background) was quantified with an Imaging-PAM system (Walz®, Effeltrich, Germany) in a light induction experiment (185 μmol photons m$^{-2}$ s$^{-1}$). Effective quantum yields of PSII [Y(II)] and levels of non-photochemical quenching [Y(NPQ)] are displayed on a false-color scale. Data were obtained from 5 leaves of 5 plants (*n* = 25) per genotype. Horizontal lines represent the medians, and boxes indicate the 25th and 75th percentiles. Whiskers denote 1.5x the interquartile range, outliers are represented by dots. Data points are plotted as circles. For statistical analyses, the non-parametric Kruskal–Wallis test was used, followed by pairwise Dunn's tests. The *p*-values were adjusted on an experiment level using the Benjamini–Hochberg method. *P*-values for comparison of complemented lines with *atcbbyab* are indicated and $p \leq 0.05$ are marked in bold.

activity. Two observations support this idea: (i) several CbbY signature and catalytic residues are specific to AtCbbYA and not found in AtCbbYB (Fig. 1b) and (ii) its catalytic efficiency on XuBP is lower than that of AtCbbYA (Table 1). A comparable case may be advanced for CA1Pase, which dephosphorylates both CA1P and PDBP[19]. This catalytic promiscuity was revealed upon the addition of recombinant CA1Pase to in-vitro assays, which prevented PDBP-dependent fallover of Rubisco activity.

Despite the reduction in carbon assimilation rates, the *atcbbyab* phenotype did not result in lethality (Fig. 2). This suggests the existence of an additional process responsible for XuBP removal in the absence of AtCbbYA and AtCbbYB. One possible explanation is that

Rubisco carboxylation of XuBP[26] is sufficient to maintain its residual enzymatic activity required for photoautotrophic growth. Alternatively, an unidentified phosphatase or enzyme could exist that recycles XuBP. In future studies, the simultaneous determination of Rubisco and Rubisco inhibitor concentrations in the chloroplasts of the different mutant lines will provide a deeper understanding of CbbY substrate specificity and the extent of Rubisco inactivation.

The restoration of wild-type-like photosynthesis observed upon targeting of the bacterial XuBP phosphatase RsCbbY to chloroplasts revealed that CbbY enzymes represent an ancient metabolite-repair system (Fig. 6). Compared to only a minor reduction in specific Rubisco activity of 10% in the *R. sphaeroides* strain without RsCbbY and

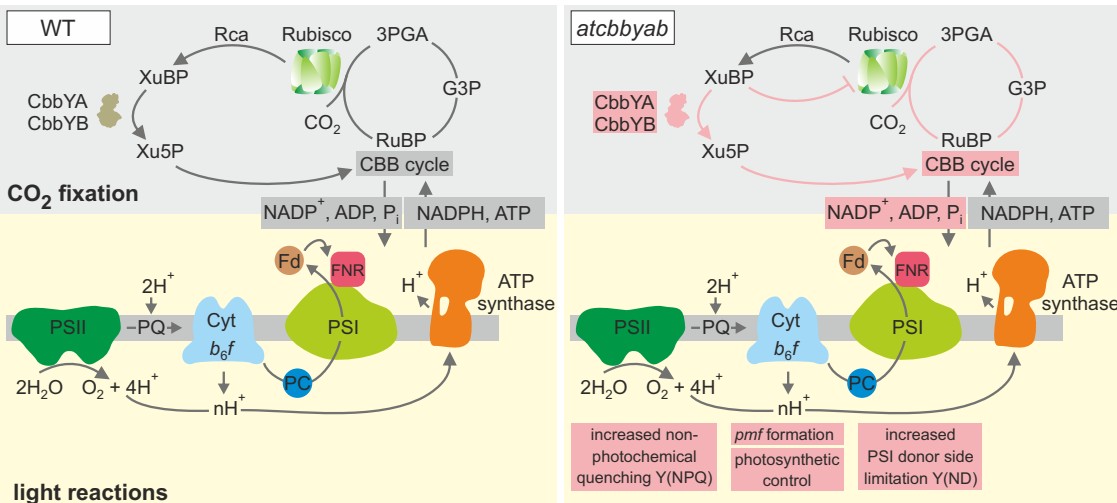

**Fig. 7 | Dephosphorylation of xylulose 1,5-bisphosphate (XuBP) to xylulose-5-phosphate (Xu5P) by CbbY phosphatases prevents downregulation of the CBB cycle and photosynthetic activity.** The Rubisco inhibitor XuBP is efficiently converted into xylulose-5-phosphate by functional cooperation between Rubisco activase (Rca) and the two XuBP phosphatases CbbYA and CbbYB in the wild type (WT, left panel). The absence of CbbY phosphatases in the *atcbbyab* double mutant results in Rubisco inhibition, downregulation of CBB cycle activity and the onset of photosynthetic control (right panel) including enhanced non-photochemical quenching [Y(NPQ)] and increased donor-side limitation of PSI [Y(ND)]. Xylulose-5-phosphate (Xu5P), 3-phosphoglycerate (3PGA), glyceraldehyde 3-phosphate (G3P), Ribulose 1,5-bisphosphate (RuBP).

the phosphoglycolate phosphatase RsCbbZ[41], assimilation rates dropped considerably to about 50% of wild-type levels in *atcbbyab* (Fig. 4). One explanation for this discrepancy is that purple bacteria and plants exhibit different kinetics in Rubisco misfiring and repair enzyme reactions. Another possibility is that purple bacteria have alternative, unknown mechanisms to remove both XuBP and 2-phosphoglycolate. In addition, XuBP and other inhibitory sugar phosphates derived from Rubisco's side reactions might be part of as yet unknown, plant-specific regulatory networks[42] that orchestrate the downregulation of photosynthetic electron transport and CBB cycle activity.

In conclusion, the discovery of the physiological significance of instances of 'misfiring' by Rubisco also has implications for efforts to increase $CO_2$ fixation efficiency in photosynthetic organisms. Since XuBP is inherently produced by Rubisco and binds directly to its active site, Rubisco encapsulation strategies, such as engineering a pyrenoid-based $CO_2$-concentrating mechanism[43,44] or cyanobacterial carboxysomes[45] in crops, need to be re-evaluated in terms of efficient release and degradation of Rubisco inhibitors in synthetic organellar subcompartments.

## Methods

### Growth conditions and plant material

Plants were grown in potting soil (A210, Stender AG, Schermbeck, Germany) exposed to a light regime of 80–120 μmol photons $m^{-2}$ $s^{-1}$ (12/12 h, light/dark) under temperature-controlled conditions (22 °C in the light phase, 18 °C in the dark phase). Plants used for crossing and seed propagation were kept under greenhouse conditions on a 14/10 h light/dark cycle, and an additional light source (HQI Powerstar 400 W/D, Osram) was used to provide supplemental illumination, resulting in a daytime light intensity of 180 μmol photons $m^{-2}$ $s^{-1}$. Wuxal Super fertilizer (8% nitrogen, 8% phosphorus, and 6% potassium; MANNA Wilhelm Haug GmbH & Co. KG, Ammerbuch-Pfäffingen, Germany) was used according to the manufacturer's instructions. High light treatment was carried out at a light intensity of 300–400 μmol photons $m^{-2}$ $s^{-1}$ (12/12 h, light/dark) under a high-pressure mercury-vapor lamp (Powerstar, HQI-T 1000 W/D, Osram, Germany).

The single mutants *atcbbya* (SALK_025204, insertion in *AT3G48420*) and *atcbbyb* (SM_3.15346, insertion in *AT4G39970*) were obtained from the SALK[46] and SLAT (John Innes Centre) collections[47], respectively. The double mutant *atcbbyab* was generated by crossing the single mutants and homozygous plants were screened by genotyping plants from the segregating $T_2$ generation. DNA was extracted as described previously[48] and combinations of gene- and T-DNA-specific primers were employed in PCR assays (see Supplementary Table 3 for primer information). Generation and characterization of the Rubisco knockdown line *rbcs1a3b-1* was described previously[32].

The *AtCbbYA-eGFP* and *AtCbbYB-eGFP* constructs used in overexpression experiments were assembled as follows. Coding regions (gene models *AT3G48420.1* and *AT4G39970.1* according to TAIR) were cloned into the binary Gateway destination vector pB7FWG2[49], thus placing them under the control of the 35S promoter. For cross-species complementation, the codon usage of the *RscbbY* gene (GenBank: U67781.1) was optimized for efficient expression in Arabidopsis, synthesized as a gBlock® (IDT®) and fused to the sequence coding for the predicted plastid transit peptide (TP) of AtCbbYA (1-74 aa). The fusion was subsequently cloned into the Gateway vector pB7FWG2, which placed *TP-RsCbbY* under the control of the CaMV 35S promoter and fused it with the eGFP-encoding reporter gene. All constructs were transformed into *Agrobacterium tumefaciens* strain GV3101, and then into *atcbbyab* plants by the floral-dip method[50]. Positive transformants were selected by restored plant growth, the presence of the GFP reporter system, and immunodetection of the fusion proteins. For each transformation, two independent, homozygous lines were isolated from the $T_3$ generation and subjected to biochemical and physiological analyses.

### Chlorophyll *a* fluorescence, P700 and gas exchange measurements

Chlorophyll *a* fluorescence and P700 parameters were determined with the Dual-PAM system (Walz, Effeltrich, Germany). The measurements shown in Fig. 3 were carried out on non-fertilized plants grown under long-day conditions (16/8 h, light/dark) for 4 weeks. Plants were dark-incubated for 20 min and then subjected to a light induction curve protocol at a light intensity of 196 μmol photons $m^{-2}$ $s^{-1}$ for 10 min. The maximum quantum yield of PSII [Fv/Fm = (Fm − Fo) / Fm)] was determined from fluorescence parameters (Fo and Fm), which were derived from the first saturating light pulse after dark incubation.

Then, saturating light pulses were applied every 30 s and photosynthetic parameters were calculated by the Dual-PAM software based on equations previously described[51,52]. Photosynthetic parameters of complemented lines were determined with an Imaging-PAM system (Walz, Effeltrich, Germany). Plants grown without fertilizer on a 12/12 h light/dark cycle (80–120 µmol photons $m^{-2}$ $s^{-1}$) for 4-5 weeks were dark-incubated for 20 min. Saturating light pulses were applied every 20 s over 10 min in a light induction experiment (185 µmol photons $m^{-2}$ $s^{-1}$) to determine fluorescence parameters in the light (Fm' and F). The effective quantum yield of PSII [Y(II)] and the yield of non-photochemical quenching [Y(NPQ)] were calculated by the equations Y(II) = (Fm' - F)/Fm' and Y(NPQ) = F/Fm' - F/Fm, respectively[52].

Rates of gas exchange were determined simultaneously with measurements of chlorophyll $a$ fluorescence parameters using a GFS-3000 system (Walz, Effeltrich, Germany) equipped with an Arabidopsis Chamber (3010-A, Walz, Effeltrich, Germany). Col-0, atcbbya and atcbbyb were analyzed after 4 weeks of growth on a 12/12 h light/dark cycle (80–120 µmol photons $m^{-2}$ $s^{-1}$). The genotypes atcbbyab and rbcs1a3b-1 were grown for 2 and 3 weeks longer, respectively, allowing them to reach developmental stages comparable to Col-0, atcbbya and atcbbyb plants. First, assimilation rates and chlorophyll fluorescence parameters were recorded at six different light intensities (0, 50, 100, 300, 600, and 1200 µmol photons $m^{-2}$ $s^{-1}$) at a constant $CO_2$ concentration (46 Pa). Subsequent measurements were then carried out at $CO_2$ pressures of 5, 20, 91 and 122 Pa. $CO_2$ assimilation rates were expressed relative to rosette leaf fresh weight (µmol $CO_2$ $h^{-1}$ $g^{-1}$ FW), which was determined for each plant after the gas-exchange measurement. The intercellular $CO_2$ mole fraction $pCi$ was determined and calculated according to Caemmerer and Farquhar[53].

## Pigment analyses

Pigments of rosette leaves were isolated from four-week-old plants grown on a 12/12 h light/dark cycle (80–120 µmol photons $m^{-2}$ $s^{-1}$). Leaves were harvested 3 h after the shift from the dark into the light and immediately homogenized in liquid nitrogen. Pigments were extracted with 99.5% acetone and separated from leaf debris by centrifugation (16,000 g) for 20 min at 4 °C. Pigments were separated by reverse-phase HPLC using a LiChrospher 100 RP-18 column with a particle size of 5 µm (Merck, Darmstadt, Germany)[54]. Solvent A consisted of a mixture of acetonitrile, methanol, and 0.1 M Tris/NaOH (pH 8.0), in a ratio of 87:10:3. Solvent B contained methanol and hexane in a ratio of 4:1. The gradient from solvent A to solvent B was developed from 9 to 12.5 min at a flow rate of 2 mL/min and eluted pigments were detected at 440 nm. The concentration of the pigments was calculated from the integrated peak area using conversion factors determined by calibration with pure pigments.

## Northern analyses

Total RNA was isolated with the aid of the TRIzol reagent (Invitrogen) from Col-0, atcbbya, atcbbyb and atcbbyab leaves that had been frozen and homogenized in liquid nitrogen. Northern analyses were carried out according to standard protocols[55]. Samples equivalent to 10 µg of total RNA were size-fractionated by electrophoresis in formaldehyde-containing agarose gels (1.2%). RNA was transferred overnight onto nylon membranes (Hybond-N+, Amersham Bioscience) by capillary transfer and subsequently fixed by UV radiation (Stratalinker® UV Crosslinker 1800). Ribosomal RNAs on nylon membranes were quantified by staining with methylene blue solution (0.02% [w/v] methylene blue, 0.3 M sodium acetate pH 5.5) to control for equal loading. Amplicons derived from cDNA were labeled with radioactive [α-$^{32}$P] dCTP (Hartmann Analytic, Braunschweig, Germany) and subsequently used as probes for the detection of AT3G48420 and AT4G39970 transcripts in hybridization experiments (see Supplementary Table 3 for primer information). Signals were recorded with the Typhoon Phosphor Imager System (GE Healthcare).

## Immunoblot analyses, protein purification, and antibody generation

Rosette leaves were harvested, frozen in liquid nitrogen, and homogenized in loading buffer (100 mM Tris-HCl, pH 6.8, 50 mM DTT, 8% [w/v] SDS, 24% [w/v] glycerol, and 0.02% [w/v] bromophenol blue). Proteins were denatured for 5 min at 70 °C. After centrifugation (5 min, 12,000 g, room temperature), solubilized proteins in the supernatant were fractionated on Tricine-SDS-PAGE (10% gels)[56]. Loading was adjusted to leaf fresh weight. Protein transfer to PVDF membranes (Immobilon-P, Millipore) was conducted using a semidry blotting system (Bio-Rad) following the supplier's instructions. PVDF membranes were stained with a Coomassie brilliant blue G-250 solution[56] to control for efficient transfer and sample loading. PVDF membranes were subsequently blocked with TBST buffer (10 mM Tris-HCl, pH 8, 150 mM NaCl, and 0.1% [v/v] Tween 20) supplemented with 3% (w/v) skim milk powder. Then, membranes were incubated overnight at 4 °C in blocking buffer containing primary antibodies. After washing three times with TBST buffer, membranes were treated with secondary antibodies in TBST supplemented with 3% (w/v) skim milk powder. After three more washes with TBST, signals were detected by enhanced chemiluminescence (Pierce, Thermo Scientific) using an enhanced chemiluminescence reader system (Fusion FX7, PeqLab).

For protein purification, AtCbbYA and AtCbbYB were heterologously expressed in BL21 (DE3) E. coli cells. Sequences encoding AtCbbYA$_{66-319aa}$ and AtCbbYB$_{47-316aa}$ were cloned into the expression vectors pET151 D-TOPO (ThermoFisher) and pET101 D-TOPO (ThermoFisher), respectively, resulting in 6xHis N-terminally tagged AtCbbYA$_{66-319aa}$ and 6xHis C-terminally tagged AtCbbYB$_{47-316aa}$ fusions, which were subsequently purified under native conditions using nickel-nitrilotriacetic acid agarose (Qiagen) following the manufacturer's instructions. For enzyme assays and antibody purification, AtCbbYA$_{66-319aa}$ and AtCbbYB$_{47-316aa}$ were fused to the maltose-binding protein (MBP). To this end, coding sequences were cloned into the pMal-c5x vector (New England Biolabs, NEB, Ipswich, MA, USA) and transformed into BL21 (DE3) E. coli cells. After heterologous expression, MBP-AtCbbYA and MBP-AtCbbYB were purified using an amylose resin (New England Biolabs) in accordance with the manufacturer's instructions.

For antibody production, 6xHis-AtCbbYA$_{66-319aa}$ and AtCbbYB$_{47-316aa}$-6xHis were injected into rabbits. The 6xHis-AtCbbYA$_{66-319aa}$ antibody was generated in Roberto Barbato's laboratory. AtCbbYB$_{47-316aa}$-6xHis antibodies were generated by Pineda (Berlin, Germany) and subsequently purified by affinity chromatography with immobilized MBP-AtCbbYB. Detection of the RsCbbY-eGFP fusion was performed with a commercially available GFP-specific antibody (A-6455, ThermoFisher Scientific) at a dilution of 1:7500. Rubisco activase (AS10 700, dilution of 1:5000) and the large subunit of Rubisco RbcL (AS03 037, dilution of 1:5000) were immunodetected with antibodies obtained from Agrisera. The secondary antibody (Sigma-Aldrich A9169, Goat Anti-Rabbit, IgG Antibody, HRP-conjugate) was used in a 1:30,000 dilution.

## Enzyme kinetics

Xylulose-1,5-bisphosphate (XuBP) was synthesized by enzyme-catalyzed aldol addition of dihydroxyacetone phosphate to glycolaldehyde phosphate[20]. The purity of XuBP-Li$_2$ was estimated by MBP-CbbYA-mediated phosphate release. Ribulose-1,5-bisphosphate (RuBP), xylulose-5-phosphate (Xu5P) and ribulose-1,5 phosphate (Ru5P) were purchased from Merck (Darmstadt, Germany). MBP-AtCbbYA and MBP-AtCbbYB were isolated as described above and were employed for enzyme activity assays, because they were of higher purity than the non-fused proteins. Calf intestinal phosphatase (CIP) was obtained from New England Biolabs and served as a positive control. Enzyme activity was assessed by determining the release of inorganic phosphate from sugar phosphates using the colorimetric

malachite green assay[57]. Malachite Green was purchased from Merck. Assays were carried out in a phosphate-free buffer (MOPS-KOH pH 7.5, 100 mM KCl, 10 mM MgCl$_2$) in the presence of the CbbY cofactor Mg$^{2+}$ at 25 °C[20].

## Subcellular localization analyses

Chloroplasts were isolated on a two-step Percoll gradient[58]. Leaves (20 g) from 4-week-old and dark-adapted wildtype plants were homogenized in a solution containing 450 mM sorbitol, 20 mM Tricine-KOH (pH of 8.4), 10 mM EDTA, 10 mM NaHCO$_3$, and 0.1% (w/v) bovine serum albumin (BSA). The homogenate was filtered using two layers of Miracloth (Calbiochem) and then centrifuged (500 g) for 6 min at 4 °C. The sedimented chloroplasts were suspended in a solution containing 800 μL of 0.3 M sorbitol, 20 mM Tricine-KOH (pH of 8.4), 2.5 mM EDTA, and 5 mM MgCl$_2$, placed on top of a two-step Percoll gradient (40–80% [v/v]) and centrifuged (6500 g) for 20 min at 4 °C. Intact chloroplasts from the interface were collected and ruptured in a buffer containing 20 mM HEPES (pH 7.5, adjusted with KOH), and 10 mM EDTA. After 30 min on ice, insoluble proteins were separated from the soluble fraction by centrifugation at 35,000 g for 30 min at 4 °C. Mitochondria were isolated as described previously[59]. Successful fractionation was verified by immunodetection of the following marker proteins: cytochrome oxidase subunit II (CoxII) for mitochondria (Agrisera AS04 053 A), LHCII type III chlorophyll a/b-binding protein (Lhcb3) for the insoluble chloroplast fraction (Agrisera, AS01 002), and Csp41b for the soluble chloroplast fraction (the Csp41b antibody was kindly provided by David Stern).

For protoplast isolation, leaves from 3- to 5-week-old plants ($P_{35S}$:AtCbbYA-eGFP, $P_{35S}$:AtCbbYB-eGFP and $P_{35S}$:RsCbbY-eGFP) were cut into strips, vacuum-infiltrated with enzyme solution (2.6 mg/mL macerozyme, 10 mg/mL cellulase, 10 mM MES pH 5.7, 20 mM KCl, 0.5 M mannitol, 10 mMCaCl$_2$, and 1 mg/mL BSA) and incubated for 3-4 h in the dark at room temperature on a shaker (50 rpm). Protoplasts were then filtered through a nylon membrane (72 μm pore size), pelleted by centrifugation (50 × g, 10 min), resuspended in 8.5 mL of a saccharose-containing buffer (10 mM MES pH 5.7, 20 mM MgCl$_2$, and 120 mg/mL saccharose) and overlaid with 2 mL of a mannitol-containing buffer (10 mM MES pH 5.7, 10 mM MgCl$_2$, 10 mM MgSO$_4$, and 0.5 M mannitol). Protoplasts found at the interphase after centrifugation (50 × g, 10 min) were washed once and then resuspended in the overlay buffer. Chlorophyll autofluorescence and eGFP signals were detected using an Axio Imager fluorescence microscope (Zeiss).

## Bioinformatic and statistical analyses

Co-expression studies of *AtCBBYA* (*AT3G48420*) were done with ATTED[28] and functional profiling of co-expressed genes was analyzed by g:Profiler[27]. Transit peptides were predicted by TargetP[60]. Prediction of AtCbbYB structure was conducted by AlphaFold[31]. Protein sequence alignments and the phylogenetic tree were constructed with the CLC workbench software (version 20). The evolutionary history of AtCbbYA, AtCbbYB and RsCbbY was inferred from sequence comparisons using the maximum-likelihood method[61]. The tree was drawn to scale, with branch lengths measured in the number of substitutions per site according to the Whelan and Goldman method[62]. The phylogenetic tree involved 24 amino-acid sequences (sequence identifiers are found in Supplementary Table 1) and a bootstrap analysis with 1000 replicates.

Boxplots were drawn with the web-tool BoxPlotR[63]. Statistical analyses were performed in *R* v3.5.2 (https://www.r-project.org/). Nonparametric Kruskal-Wallis tests were used, which were followed by pairwise Dunn's tests employing the *R* package *dunn.test*. P-values were adjusted on an experiment level using the Benjamini–Hochberg method.

## Accession numbers

AtCbbYA (At3g48420), AtCbbYB (At4g39970), RsCbbY (UniProt ID P95649).

## Reporting summary

Further information on research design is available in the Nature Portfolio Reporting Summary linked to this article.

## Data availability

The authors declare that all data presented in this study are available in the figures and the accompanying Supplementary Information file. The source data underlying Fig. 2 b–f; Fig. 3; Fig. 4; Fig. 5; Fig. 6a, c; Supplementary Fig. 2b, c; Supplementary Fig. 3b; Supplementary Fig. 4; Supplementary Fig. 5; and Supplementary Table 2, as well as detailed corresponding statistics, are provided as a Source Data file. Other data that support the study are available from the corresponding author. Source data are provided with this paper.

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

## Acknowledgements

We thank Paul Hardy for critical reading of the manuscript, the German Research Foundation (DFG, grant TRR175 to D.L., grant RU 1945/2–1 and RU 1945/4-1 to T.R.) for financial support, Silke Ruberg and Lea Rosenhammer for technical support, Hiroyuki Ishida for *rbcs1a3b-1* seeds, Ute Armbruster for *atcbbya* (SALK_025204) seeds, and David Stern for providing the CSP41b antibody.

## Author contributions

T.R. and D.L. designed experiments. Experiments in Arabidopsis were performed by T.R., N.K., B.R., V.P., and S.B., A.S. and M.P. synthesized XuBP and carried out enzyme kinetics studies. T.N. designed and performed gas exchange experiments. P.J. analyzed pigment composition by HPLC and R.B. generated the AtCbbYA antibody. T.R. was responsible for conceptualization and management of the entire study and wrote the paper with contributions from all authors.

## Funding

## Competing interests

The authors declare no competing interests.
