## [Peer Review File · Nature Communications]

An ancient metabolite damage-repair system sustains photosynthesis in plantsReviewer #1 (Remarks to the Author):

Recently it has become apparent that metabolism is highly prone to damage by misfire reactions. Enzymes are not completely specific, but will catalyze off-pathway reactions at various rates. There are then additional enzymes required to correct these mistakes, and remove the deleterious products. The main photosynthetic carboxylase Rubisco has long been known to feature one of the most elaborate repair schemes, known as photorespiration, where 2-PG is repaired by recycling it back to 3-PG. In addition it has long been known that the enzyme generates misfire products that potentially inhibit its own function. Interestingly there has been very little work done to understand the physiological consequence of these off-pathway reactions.

The paper by Leister and colleagues is a very important contribution that describes a strong photosynthetic phenotype for plants that are missing two homologues of the XuBP phosphatase identified by Bracher et al. some years ago. As expected for an unchecked build-up of a Rubisco inhibitor, the phenotype is similar to that of a Rubisco knockdown. The manuscript is already highly polished, and very carefully describes its findings. It will be a landmark study for both CO₂ assimilation/photosynthesis and the expanding field of metabolite repair.

It was an extremely satisfying conclusion to see that the XuBP phosphatase from a photosynthetic bacterium can completely rescue the higher plant lesion.

The experiments are very well done and carefully performed. I mostly have some suggestions for the discussion to raise some open questions that could be addressed in the next study:

1. It would be very interesting to determine the concentrations of XuBP in the chloroplast stroma of the different lines. As the phenotype is not lethal, and clearly Rubisco is functional to a reduced extent, there must be another yet unidentified phosphatase (or other enzyme) removing XuBP? Of course this could be even a side reaction of a CBB-pathway enzyme like FBPase?

Minor points:

1. L45: Consider updating/supplementing the classic Ellis with the newer, more sophisticated analysis by Bar-On and Milo
2. L417: the lack of a strong phenotype of RsCBBY and RsCbbZ is not really that interesting, it just indicates that (like in Arabidopsis) there are additional phosphatases present R. sphaeroides that have not been hit in the cited study..
"it is reasonable to assume that the longer lifetimes of plant cells... enhance accumulation of deleterious Rubisco inhibitors" I cannot follow this argument, I would argue the accumulation of deleterious Rubisco inhibitors is completely independent of cell lifetime, but depends on the kinetics of both Rubisco's misfire reactions and the quantity and kinetics of the repair enzymes. Again, at this stage, the best explanation is that R. sphaeroides has alternative ways of dealing with both XuBP and 2-PG.
3. Figure 5b: The data shown for RuBP phosphatase makes it impossible to recapitulate the kinetics reported in Table 1. Please insert a relevant Figure into supplementary (with relevant y-axis)

Fix details of ref. 33, 41

Reviewer #2 (Remarks to the Author):

Leister et al demonstrate the activity of XuBPase (CbbY) is required for efficient photosynthetic activity in plants. They show that, as in other species, two genes are present in Arabidopsis and knocking out both genes has deleterious effects for photosynthesis and plant growth without

impacting the abundance of Rubisco or Rubisco activase. Their findings support a role of these phosphatases as a metabolite damage-repair system. The experiments are well thought out and conducted and the research advances knowledge in an area relatively under-explored to date.

I have only minor comments for the authors to consider:

Line 46 – whether RuBP carboxylation is the rate limiting step in the CBB depends on the species and prevailing conditions, consider revising.

Line 53 – some colleagues would argue that photorespiration is useful, rewording might be adequate.

Line 61 – for the benefit of readers less familiar with Rubisco, it would be useful to introduce carbamylation of catalytic sites prior to this sentence.

Fig 2 – state the light level used for the different light regimes and explain whether the p-values are for comparison of each line with Col-0. The latter applies to other similar figures with statistical analysis of data.

Fig 4 – could the units of net CO₂ assimilation be converted to $\mu\text{mol m}^{-2} \text{s}^{-1}$?

Line 543 – provide information on the secondary antibodies used.

Supp Table S2 – could this dataset be supported by statistical analysis?

Antibody production – in future research, please consider using recombinant antibody production instead of animal-based.

To the Editors of *Nature Communications* and reviewers of the manuscript NCOMMS-23-06969A

Point-by-point response to the reviewers' comments

Reviewer #1 (Remarks to the Author):

Recently it has become apparent that metabolism is highly prone to damage by misfire reactions. Enzymes are not completely specific, but will catalyze off-pathway reactions at various rates. There are then additional enzymes required to correct these mistakes, and remove the deleterious products. The main photosynthetic carboxylase Rubisco has long been known to feature one of the most elaborate repair schemes, known as photorespiration, where 2-PG is repaired by recycling it back to 3-PG. In addition it has long been known that the enzyme generates misfire products that potently inhibit its own function. Interestingly there has been very little work done to understand the physiological consequence of these off-pathway reactions.

The paper by Leister and colleagues is a very important contribution that describes a strong photosynthetic phenotype for plants that are missing two homologues of the XuBP phosphatase identified by Bracher et al. some years ago. As expected for an unchecked build-up of a Rubisco inhibitor, the phenotype is similar to that of a Rubisco knockdown.

The manuscript is already highly polished, and very carefully describes its findings. It will be a landmark study for both CO₂ assimilation/photosynthesis and the expanding field of metabolite repair.

It was an extremely satisfying conclusion to see that the XuBP phosphatase from a photosynthetic bacterium can completely rescue the higher plant lesion.

The experiments are very well done and carefully performed. I mostly have some suggestions for the discussion to raise some open questions that could be addressed in the next study:

1. It would be very interesting to determine the concentrations of XuBP in the chloroplast stroma of the different lines. As the phenotype is not lethal, and clearly Rubisco is functional to a reduced extent, there must be another yet unidentified phosphatase (or other enzyme) removing XuBP? Of course this could be even a side reaction of a CBB-pathway enzyme like FBPase?

Response: We would like to thank the reviewer for the suggestion regarding the discussion section. We agree that determining XuBP levels in the mutant collection will provide deeper insights into the substrate specificity of AtCbbY enzymes and the extent of Rubisco inactivation *in planta*. We plan to include this analysis in an upcoming study. The following paragraph has been added to the discussion section:

“Despite the reduction in carbon assimilation rates, the *atcbbyab* phenotype did not result in lethality (Fig. 2). This suggests the existence of an additional process responsible for XuBP removal in the absence of AtCbbYA and AtCbbYB. One possible explanation is that Rubisco carboxylation of XuBP²⁶ is sufficient to maintain its residual enzymatic activity required for photoautotrophic growth. Alternatively, an unidentified phosphatase or enzyme could exist that recycles XuBP. In future studies, the simultaneous determination of Rubisco and Rubisco inhibitor concentrations in the chloroplasts of the different mutant lines will provide a deeper understanding of CbbY substrate specificity and the extent of Rubisco inactivation.”

Minor points:

1. L45: Consider updating/supplementing the classic Ellis with the newer, more sophisticated analysis by Bar-On and Milo

Response: We replaced the former reference by Bar-On and Milo (2019).

2. L417: the lack of a strong phenotype of RsCBBY and RsCbbZ is not really that interesting, it just indicates that (like in Arabidopsis) there are additional phosphatases present R. sphaeroides that have not been hit in the cited study..

"it is reasonable to assume that the longer lifetimes of plant cells... enhance accumulation of deleterious Rubisco inhibitors" I cannot follow this argument, I would argue the accumulation of deleterious Rubisco inhibitors is completely independent of

cell lifetime, but depends on the kinetics of both Rubisco's misfire reactions and the quantity and kinetics of the repair enzymes. Again, at this stage, the best explanation is that *R. sphaeroides* has alternative ways of dealing with both XuBP and 2-PG.

Response: We have rephrased the paragraph and will now discuss the differences between Rhodobacter and Arabidopsis as proposed by the reviewer:

“Compared to only a minor reduction in specific Rubisco activity of 10% in the *R. sphaeroides* strain without RsCbbY and the phosphoglycolate phosphatase RsCbbZ⁴¹, assimilation rates dropped considerably to about 50% of wild-type levels in *atcbbyab* (Fig. 4). One possibility for this discrepancy is that purple bacteria and plants exhibit different kinetics in Rubisco misfiring and repair enzyme reactions. Another explanation is that purple bacteria have alternative, unknown mechanisms to remove both XuBP and 2-phosphoglycolate.”

3. Figure 5b: The data shown for RuBP phosphatase makes it impossible to recapitulate the kinetics reported in Table 1. Please insert a relevant Figure into supplementary (with relevant y-axis)

Response: We now show CbbY kinetics with RuBP as substrate and a fitted scaling of the y-axis in Supplementary Figure 5b. We also refer to Supplementary Figure 5b in the figure legend of Fig. 5: “Note that a plot of CbbY kinetics with RuBP as substrate and a fitted scaling of the y-axis is provided in Supplementary Figure 5b.”

Fix details of ref. 33, 41

Response: Both references are now corrected.

Reviewer #2 (Remarks to the Author):

Leister et al demonstrate the activity of XuBPase (CbbY) is required for efficient photosynthetic activity in plants. They show that, as in other species, two genes are present in Arabidopsis and knocking out both genes has deleterious effects for photosynthesis and plant growth without impacting the abundance of Rubisco or Rubisco activase. Their findings support a role of these phosphatases as a metabolite damage-

repair system. The experiments are well thought out and conducted and the research advances knowledge in an area relatively under-explored to date.

I have only minor comments for the authors to consider:

Line 46 – whether RuBP carboxylation is the rate limiting step in the CBB depends on the species and prevailing conditions, consider revising.

Response: We revised the sentence.

“Carboxylation of ribulose-1,5-bisphosphate (RuBP) is the first step in the Calvin-Benson-Bassham (CBB) cycle,...”

Line 53 – some colleagues would argue that photorespiration is useful, rewording might be adequate.

Response: We reworded the paragraph.

“Despite its central role in the CBB cycle, Rubisco has a low turnover rate of 3-10 CO₂ molecules per sec⁴ and its complex reaction mechanism is prone to error. One of its by-products – 2-phosphoglycolate⁵ – effectively inhibits several enzymes of primary carbon metabolism in photosynthetic organisms, including triosephosphate isomerase⁶, phosphofructokinase⁷ and sedoheptulose 1,7-bisphosphate phosphatase⁸. In addition, 2-phosphoglycolate is recycled in a metabolic process called photorespiration which requires ATP and multiple enzymes, and involves no less than four subcellular compartments⁹. Another limitation for efficient carboxylation is that protonation and oxygenation of the RuBP enediolate intermediate gives rise to several isomeric pentulose bisphosphates [2,3-pentodiulose-1,5-bisphosphate (PDBP), 3-ketoarabinitol-1,5-bisphosphate and xylulose-1,5-bisphosphate (XuBP)]¹⁰, which are also produced during *in-vitro* studies with isolated Rubisco complexes.”

Line 61 – for the benefit of readers less familiar with Rubisco, it would be useful to introduce carbamylation of catalytic sites prior to this sentence.

Response: A paragraph about Rubisco activation has been included.

“Carboxylation of ribulose-1,5-bisphosphate (RuBP) is the first step in the Calvin-Benson-Bassham (CBB) cycle, which requires the activation of Rubisco through carbamylation of a conserved lysine residue in the active site by a non-substrate

CO₂ molecule³. The carbamylated lysine is then stabilized by the subsequent binding of a magnesium ion, which enables the efficient electrophilic attack of RuBP by the substrate CO₂ molecule.”

Fig 2 – state the light level used for the different light regimes and explain whether the p-values are for comparison of each line with Col-0. The latter applies to other similar figures with statistical analysis of data.

Response: We now state light levels and provide a clearer explanation of p-value representations in all relevant figures as recommended by the reviewer.

Fig 4 – could the units of net CO₂ assimilation be converted to $\mu\text{mol m}^{-2} \text{s}^{-1}$?

Response: We analysed carbon assimilation rates of whole rosettes in an Arabidopsis Chamber (3010-A, Walz, Effeltrich, Germany) and related the rates to the fresh weight of the rosette leaves. We did not determine the plant area, so the units of net CO₂ assimilation cannot be converted to $\mu\text{mol m}^{-2} \text{s}^{-1}$.

Line 543 – provide information on the secondary antibodies used.

Response: Information about the secondary antibody is now provided. “The secondary antibody (Sigma-Aldrich A9169, Goat Anti-Rabbit, IgG Antibody, HRP-conjugate) was used in a 1:30,000 dilution.”

Supp Table S2 – could this dataset be supported by statistical analysis?

Response: A statistical analysis has been added to Supplementary Table 2.

Antibody production – in future research, please consider using recombinant antibody production instead of animal-based.

Response: We will consider recombinant antibody production in future research projects.